# Controlling Nitrogen Removal Processes in Improved Vertical Flow Constructed Wetland with Hydroponic Materials: Effect of Influent COD/N Ratios

Elackiya Sithamparanathan, Nora B. Sutton, Huub H. M. Rijnaarts [ID] and Katarzyna Kujawa-Roeleveld *

Department of Environmental Technology, Wageningen University & Research, P.O. Box 17, 6700 AA Wageningen, The Netherlands
* Correspondence: katarzyna.kujawa@wur.nl; Tel.: +31-317485404

**Abstract:** Discharge of nitrogen (N) with wastewater causes eutrophication in surface water. On the other hand, nutrient-rich wastewater can be valuable for agriculture. Tailoring N removal or conservation is important to meet the requirements of different water end uses. Improved vertical flow constructed wetlands with hydroponic materials (CWH) as substrata were developed at lab scale in a greenhouse and studied to optimize N removal in CWH. This study investigated the effect of influent COD/N ratios of 5/1 and 15/1 on the removal or conservation of N in CWHs with Syngonium as vegetation and three substrata, pumice, cocopeat, and mineral wool. CWH with pumice showed the highest TN removal at both COD/N ratios. The Syngonium plant significantly contributed to the additional 50% TN removal in CWH. Nitrification of above 90% was observed at both studied COD/N ratios, indicating sufficient oxygenation due to the vertical pulse flow operated CWH. The denitrification process was enhanced at a higher COD/N ratio of 15/1 compared to 5/1 by around 10–40%. The occurring nitrification and denitrification indicate the coexistence of aerobic and anaerobic conditions in CWH, and balancing these conditions is necessary for future applications to remove N for its specific end use, i.e., irrigation water (high standards) or discharge to surface water (low standards).

**Keywords:** COD/N ratio; constructed wetlands; hydroponic materials; nitrification; denitrification; wastewater reuse

## 1. Introduction

Excessive inputs of nutrients, nitrogen (N), and phosphorus (P) to aquatic environments lead to eutrophication which deteriorates water quality for aquatic life, drinking, industry, recreation or other uses [1,2]. In addition, the presence of nitrate in aquatic bodies is linked to carcinogenic potency and birth defects in humans [3]. Therefore, it is important to remove nutrients from wastewater. On the other hand, nutrients are of high value for fertigation, nutrient-rich water [1]; therefore, in the case of irrigation, nutrients should be conserved in (waste) water. Since wastewater effluents are often considered for reuse as irrigation water, tailoring N and P removal and conservation for different water end uses is important to make optimal use of the fertilizing potential of wastewater.

Constructed wetlands (CWs) are known nature-based and cost-effective post-treatment technologies that have low energy requirements compared to conventional wastewater treatment systems and therefore, a sustainable solution to (partially) remove nutrients from wastewater along with other conventional and emerging pollutants [4,5]. However, the nutrient removal efficiency of CWs is mostly insufficient, and depending on CW design parameters and environmental conditions, the removal efficiency largely varies between 50–95% for total nitrogen (TN) and 18–76% for total phosphorus (TP) [6–9]. In subsurface flow CWs with unaerated surface flow and subsurface flow, nitrogen removal is mainly limited by a poor nitrification process due to a lack of oxygen, i.e., nitrifying bacteria

compete with organics for limited oxygen. Additionally, CWs are also often affected by insufficient electron donors for biological denitrification, resulting in poor N removal of around 50% in most cases [2].

The treated water from CWs needs to meet certain effluent quality requirements for (re)use. The maximum permissible concentrations of TN and TP in treated water are regulated to be below 10 mg/L and 2 mg/L, respectively, for irrigation use in European Union [10,11]. However, treated water from CWs often exceeds these standards [5,6]; for example, average effluent TN concentrations have been reported between 21–35 mg/L in various CWs studied in Europe with the influent TN concentrations between 37–46 mg/L [12]. If treated effluent has to be discharged into surface water bodies, requirements regulated by the European Commission need to be followed: TN and TP concentrations should be below 20 mg/L and 1–2 mg/L, respectively [13]. In order to meet the requirements of reuse of treated water for irrigation or to discharge into surface water bodies, the treatment performance of CWs in terms of nutrient removal has to be improved which can be completed by various approaches.

One approach to tailoring nutrient removal is the employment of adequate hydroponic materials such as the CW filter bed substrata. Compared to conventional CWs substrata such as sand and gravel, hydroponic substrata, for example, pumice, cocopeat or mineral wool have enhanced or more beneficial physicochemical characteristics such as higher porosity, surface area, and higher water holding capacity [14,15] that may offer new opportunities to steer biological and physicochemical nutrient removal in CWs. For example, hydroponic substrata have a higher surface area which could support microbial adhesion and biofilm development. These substrata are also highly porous which could therefore allow oxygen diffusion into the filter bed enhancing the microbial activities and nitrification process. Substrata of different physiochemical properties may support the formation of anaerobic niches which could facilitate the denitrification process. Therefore, it was hypothesized that hydroponic materials can support improved and more controllable N removal in CWs. Since there are no literature references available on the effects of these substrata on nutrient removal in CWs, this study focused on the removal or conservation of N by steering nitrification and denitrification in CWs with hydroponic substrata. This study also examined TP removal in the system but did not investigate how to control TP effluent concentration. The hypothesis that hydroponic materials can be used to tailor N removal for meeting different effluent requirements was investigated by building mesocosm-improved vertical flow constructed wetland (CW) with hydroponic materials (H) such as pumice, cocopeat, and mineral wool as referred to CWH.

Biological N removal involves two major processes: nitrification and denitrification. In VFCWs, due to vertical flow and pulse feeding, sufficient atmospheric oxygen should be supplied which could facilitate the nitrification process but may prevent denitrification because of aerobic conditions. Recent studies identify certain approaches, for example, denitrifying ammonium oxidation process (DEAMOX) as an alternative to improve nitrogen removal when both $NH_4^+$-N and $NO_3^-$-N are present in wastewater simultaneously. However, these processes in biofilm reactors or CWs require further investigation [16]. Furthermore, denitrification is also often limited in CWs due to insufficient organic carbon [17]. In conventional wastewater treatment, the process is configured in such a way that optimal use is made from carbon available in wastewater influent (pre-denitrification systems with internal recycling); otherwise, an additional carbon source is sometimes added to enhance the denitrification process. There is a trade-off between COD dosing for denitrification and sufficient oxygen for nitrification. COD oxidation with oxygen could deplete the available oxygen for nitrification which could result in poor nitrification and eventually result in poor denitrification as well. Therefore, there is a need to identify the optimal amount of COD (represented as COD/N) to be maintained in the influent of CWH to facilitate both nitrification and denitrification processes [18].

Plants in CWs also influence N removal. Plant roots provide a surface for microbial adhesion and biofilm development and provide optimal pH conditions for rhizosphere

microbial communities that can facilitate both nitrification and denitrification processes. In addition, plant roots can release oxygen creating aerobic conditions to facilitate the nitrification process in CWs. On the other hand, oxygen release from plant roots could limit the denitrification process. [6]. Therefore, this study also investigated the influence of plants on overall N removal in CWH. The contribution of the plant to remove N in CWH is denoted by the term "phytoremediation" which includes various plant-associated removal processes: plant uptake, phytoaccumulation, phytodegradation or phytotransformation and adsorption to root surfaces [6,19]. The Syngonium plant was selected for CWH based on the previous study (unpublished work) where the plant significantly enhanced TN removal (by around 87%).

Based on the literature evidence, this study selected two influent COD/N ratios: 5/1 and 15/1. The COD/N of 5/1 is often reported as the optimal ratio to achieve high TN removal in CWs [20,21]. A too-low COD/N ratio (below 5/1) may lead to an insufficient denitrification [21]. Further, the increase in the COD/N ratio (e.g., to 15/1) with external carbon sources was found to be successful in increasing the overall TN removal in numerous studies in CWs [18,22,23]. Therefore, COD/N of 15/1 was selected as a good representative for the high COD/N ratio and to clearly observe the effect of low and high COD/N ratios on N removal.

This study investigated the effect of influent COD/N ratio of 5/1 and 15/1, substrata type, and plant on N removal or conservation in CWH using pumice, cocopeat, and mineral wool as CWH hydroponic substrata, and Syngonium as plant type. The results of this study are interpreted for optimizing CWH for N conservation to either produce irrigation water or water that can be discharged to surface water meeting the effluent quality requirements.

## 2. Materials and Methods

### 2.1. Substrata and Plant

Hydroponic materials for CWHs, namely, pumice and cocopeat were provided by Greenyard Horticulture Belgium NV, and mineral wool was provided by Drainblock BV, the Netherlands. The properties of the substrata are presented in Table A1. Ornamental plant *Syngonium Podophyllum* was selected for CWH. Syngonium plant was specifically selected as it is a robust plant with the ability to withstand various environmental conditions and it has the aesthetical and economic values [24]. Furthermore, Syngonium plant improved TN removal in the previous studies (unpublished work). The plants were purchased from the Garden Centre De Oude Tol BV, the Netherlands.

### 2.2. Influent Preparation

Synthetic influent for CWH was prepared with tap water using a modified Hoagland nutrient solution as the composition described by [25], containing 10 mM P buffer of $Na_2HPO_4.2H_2O$ and $KH_2PO_4$, 50 mg/L of $NH_4^+$-N (in form of $NH_4Cl$, mimicking N concentrations in municipal wastewater), and methanol as the extra source of designed COD/N ratio. In order to achieve the influent COD/N ratio of 5/1 and 15/1, either 250 mg/L or 750 mg/L of COD methanol was added. The P buffer maintained the influent pH around 7 throughout the experiment (Figure A1). Influent was prepared in a buffer tank, stirred, and maintained at 4 °C by heat exchangers to avoid bacterial or algae growth. The tank's content was refreshed every 3 days.

### 2.3. CWH Configurations and Operation

Mesocosm CWHs were built in individual glass containers of 25 cm × 25 cm × 25 cm (Figures 1 and 2). Each container was connected to 5 L effluent bottle for draining water and collecting the effluent for analysis. The glass beads were added to the bottom of each CWH container to avoid clogging the outlet. The study included 2 test groups in duplicates per substratum: (1) planted pumice or cocopeat or mineral wool with Syngonium as CWH-PS, CWH-CS, and CWH-MS, respectively; (2) pumice or cocopeat or mineral wool unplanted groups as CWH-P, CWH-C, and CWH-M, respectively. The experimental set-up was located

in the climatized glasshouse of Wageningen University and Research, the Netherlands. The greenhouse was selected in order to have a higher and more stable temperature. The daytime temperature in the greenhouse was 20 °C and it was 18 °C at night. The humidity fluctuated between 30–85%.

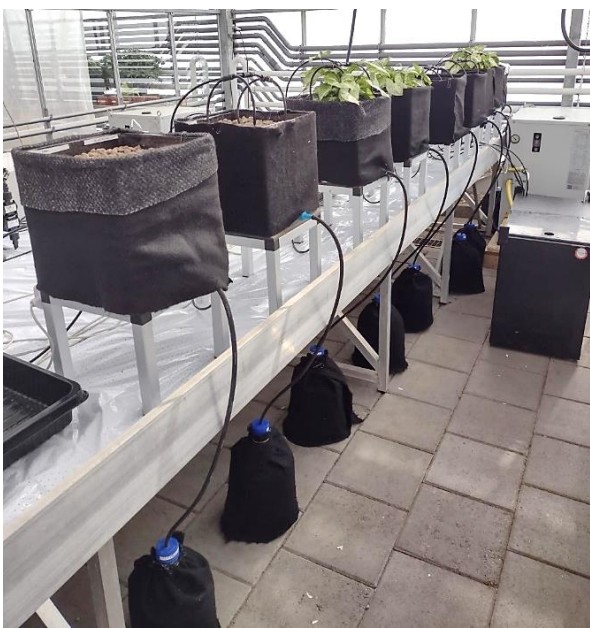

**Figure 1.** Mesocosm CWH built in the greenhouse of Wageningen University and Research, The Netherlands.

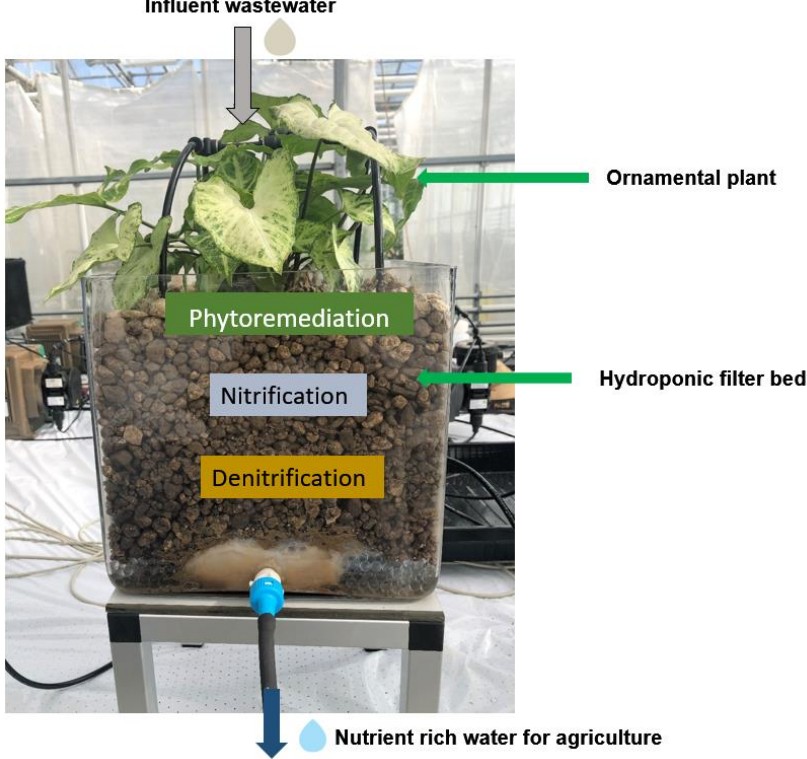

**Figure 2.** Mesocosm CWH.

The experiment was acclimatized for 113 days to establish a stable, neutral pH in all the CWH in order to ensure neutral to slightly alkaline pH for nitrification and denitrifica-

tion processes in each CWH at the beginning of the experiment [26,27]. Both during the acclimation period and main experimental period, CWH was fed with synthetic influent as described in Section 2.2. CWH was fed intermittently with 180 mL influent/minute at every 4 h of cycle. After the acclimation period, the experiment was carried out for 28 days at COD/N of 5/1 and subsequently for 28 days at COD/N of 15/1. CWH had the following characteristics: (1) HLR of 17 mm/day; (2) HRT of 2–3 days; (3) influent COD of 250 mg/L at COD/N of 5/1 and influent COD of 750 mg/L at COD/N of 15/1; (4) influent TN of 50 mg/L. During the study, the influent and effluent samples were collected once a week at the end of 5 cycles (or the composite samples of 20 h) and analyzed for pH, DO, COD, $NH_4^+$-N, $NO_3^-$N, and TN.

### 2.4. Sample Preparation and Analytical Techniques

Collected samples were centrifuged at 5000 RPM for 10 min at 20 °C and stored at 4 °C until further analysis. COD, $NH_4^+$-N, $NO_3^-$N, and TN were analyzed by Dr. Lange test kits (Hach Lange GmbH, Düsseldorf, Germany) and measured by a Hach DR/3900 spectrophotometer (Germany). The pH and DO were monitored by a multi-digital meter (HACH HQ40d, Ames, IA, USA). With regard to DO measurement, oxygen sensor spot SP-PSt3-NAU purchased from PreSens Precision Sensing GmbH, Germany was attached to the inner surface of every CWH and DO was measured contactless through the transparent glass container. The measured DO concentration in all the CWH was between 10–20 mg/L throughout the experiment and therefore, the values were above the level of saturated DO of 7.0–9.0 mg/L in water [28]. Thus, these values were considered non-reliable and not taken into account further in this study.

### 2.5. Data Analysis

The overall removal efficiency of each constituent was calculated using the following equation:

$$Removal\ efficiency\ (\%) = \frac{C_{in} - C_{out}}{C_{in}} \times 100$$

where $C_{in}$ concentration in the influent (mg/L)

$C_{out}$ concentration in the effluent (mg/L)

To solely estimate nitrification and denitrification and exclude the effect of plants on nutrient uptake, CWH without plants was considered. Therefore, the loss of $NH_4^+$-N (mg/L) can be translated into "the estimated nitrification (mg/L)" which is the difference between influent $NH_4^+$-N (mg/L) and effluent $NH_4^+$-N (mg/L). Similarly, denitrification was estimated as the difference between estimated nitrification or nitrified $NH_4^+$-N and effluent $NO_3^-$-N concentration.

A *t*-test was conducted using Microsoft Excel to test whether the observed differences in terms of nitrification and denitrification among CWH-P (unplanted pumice CWH), CWH-C (unplanted cocopeat CWH), and CWH-M (unplanted mineral wool CWH), were statistically significant or not at both COD/N ratios.

## 3. Results and Discussion

### 3.1. Removal of TN and COD in CWH

TN and COD removal were investigated in CWH with different hydroponic substrata. In the unplanted groups, CWH with pumice (CWH-P) showed higher TN removal than with cocopeat (CWH-C) and mineral wool (CWH-M) at both COD/N ratios (Table 1). On average, at a COD/N ratio of 5/1, TN removals were found for CWH-M and CWH-C between 3–7% whereas for CWH-P around 35%. Increasing the COD/N ratio to 15/1, TN removal increased for all unplanted systems to 33%, 24%, and 84% for CWH-M, CWH-C, and CWH-P, respectively. When comparing planted with unplanted groups, Syngonium significantly contributed to the overall TN removal in CWH-MS, CWH-CS, and CWH-PS;

an additional amount of TN removal was found to be around 50% at COD/N of 5/1 and of 10–40% at 15/1 ratio (Table 1).

**Table 1.** Average TN influent and effluent concentrations (mg/L) and removal (%) with standard error in CWH over the experimental period at COD/N ratio of 5/1 and 15/1, in comparison with the effluent requirements for discharge into surface water and reuse for irrigation water. Underlined values meet the effluent quality requirements. CWH-MS and CWH-M indicate mineral wool with and without plants, respectively. CWH-CS and CWH-C indicate cocopeat with and without plants, respectively. CWH-PS and CWH-P indicate pumice with and without plants, respectively.

| | COD/N = 5/1 | | COD/N = 15/1 | |
|---|---|---|---|---|
| **CWH** | **TN Concentration (mg/L)** | **TN Removal (%)** | **TN Concentration (mg/L)** | **TN Removal (%)** |
| Influent | $47 \pm 6$ | | $51 \pm 1$ | |
| Effluent: | | | | |
| CWH-M | $45 \pm 1$ | $3 \pm 1$ | $34 \pm 1$ | $33 \pm 3$ |
| CWH-MS | $22 \pm 2$ | $51 \pm 4$ | $\underline{15 \pm 2}$ | $71 \pm 5$ |
| CWH-C | $43 \pm 6$ | $7 \pm 13$ | $39 \pm 4$ | $24 \pm 8$ |
| CWH-CS | $23 \pm 7$ | $53 \pm 19$ | $30 \pm 2$ | $40 \pm 4$ |
| CWH-P | $29 \pm 4$ | $35 \pm 8$ | $\underline{8 \pm 2}$ | $84 \pm 4$ |
| CWH-PS | $\underline{6 \pm 1}$ | $86 \pm 2$ | $\underline{3 \pm 1}$ | $93 \pm 2$ |
| Effluent requirement for discharge into surface water [1] | 20 | | 20 | |
| Maximum permissible concentration for irrigation water [2] | 10 | | 10 | |

Note: [1] [13], [2] [10,11].

COD removal was high in almost all the CWHs at both COD/N ratios: more than 90% of COD was removed, except for cocopeat where the removal varied between 60–88% (Figure A2). The elevated concentrations of COD from cocopeat were earlier found to be attributed to the leaching of organic compounds from cocopeat into the effluent resulting in higher COD in the cocopeat and therefore a lower effective COD removal.

The performance of CWH was investigated by comparing the effluent concentration to two different end-use targets: (i) discharge to surface water, i.e., the European Union (EU) effluent standards for discharging the treated water into surface water [13]and (ii) reuse as irrigation water, i.e., the Food and Agriculture Organization (FAO) effluent requirements for reuse as irrigation water [10,11] (Table 1). CWH-PS met the requirement for TN (<20 mg/L) at both COD/N ratios for discharge to surface waters. At the COD/N of 15/1, CWH-MS, CWH-PS, and CWH-P met this requirement for TN. When considering the reuse requirements for irrigation water, CWH-PS met the requirement for TN (<10 mg/L) at both COD/N ratios; at the COD/N of 15/1, CWH-PS and CWH-P met the requirement for TN. In particular, CWH of pumice could produce effluent which meets effluent quality requirements for both discharging into surface waters and irrigation waters when treating wastewater with an initial TN concentration of 41–53 mg/L. This is similar to untreated or primarily treated municipal wastewater.

*3.2. Nitrogen Removal in CWH*

3.2.1. Nitrification in CWH

Nitrification efficiencies were estimated and evaluated for the studied CWH mesocosms. The efficiency reached more than 90% in all the unplanted groups for both COD/N ratios (Tables 2 and A2). Different COD/N ratios and variations in substrata type did not seem to affect the nitrification process as the efficiency values were similar under all the studied conditions. Significant nitrification was supported by a *t*-test carried out comparing the data of the two ratios (p values were higher than 0.05 and insignificantly different for all studied groups at the two ratios (Table A4). The required amount of oxygen for complete oxidation of influent $NH_4^+$-N was calculated as 218/48 mg/L oxygen required/mg/L of

$NH_4^+$-N oxidized. The influent DO concentration was always above the saturated level of DO (7.0–9.0 mg/L) in water; therefore, atmospheric oxygen influx into the CWH mesocosms must have been contributing to the almost complete oxidation of influent $NH_4^+$-N in all the CWH systems. The vertical design and pulse feeding of mesocosms CWH apparently allowed sufficient atmospheric oxygen to enter the system thus facilitating high levels of nitrification, as has been found by others [29,30].

**Table 2.** Influent and effluent $NH_4^+$-N concentration, effluent $NO_3^-$-N concentration, estimated nitrification, and denitrification (mg/L) with standard error in CWH over the experimental period at both COD/N ratios. CWH-M, CWH-C, and CWH-P indicate mineral wool, cocopeat, and pumice without plants, respectively. The denoted values of (a and a1), (b and b1), (c and c1), (a and c), (b and c), and (a1, b1, and c1) are statistically significant according to *t*-test.

| CWH | COD/N = 5/1 | | | | COD/N = 15/1 | | | |
|---|---|---|---|---|---|---|---|---|
| | [$NH_4^+$-N] (mg/L) | [$NO_3^-$-N] (mg/L) | Estimated Nitrification/ Nitrified $NH_4^+$-N (mg/L) | Estimated Denitrification/ Denitrified $NO_3^-$-N (mg/L) | [$NH_4^+$-N] (mg/L) | [$NO_3^-$-N] (mg/L) | Estimated Nitrification/ Nitrified $NH_4^+$-N (mg/L) | Estimated Denitrification/ Denitrified $NO_3^-$-N (mg/L) |
| Influent | 46.7 ± 4.2 | - | - | - | 48.7 ± 0.8 | - | - | - |
| CWH-M | 1.9 ± 0.5 | 39.5 ± 3.9 | 45 ± 4 | 5 ± 2 [a] | 0.8 ± 0.4 | 29.3 ± 2.0 | 48 ± 1 | 19 ± 2 [a1] |
| CWH-C | 2.4 ± 1.8 | 37.1 ± 7.8 | 44 ± 6 | 7 ± 3 [b] | 0.4 ± 0.3 | 35.4 ± 1.6 | 48 ± 1 | 13 ± 2 [b1] |
| CWH-P | 4.7 ± 1.1 | 20.5 ± 3.1 | 42 ± 4 | 21 ± 4 [c] | 2.8 ± 0.8 | 3.1 ± 2.3 | 46 ± 1 | 43 ± 3 [c1] |

### 3.2.2. Denitrification in CWH

The denitrification process occurred in all the CWH at both COD/N ratios but with different efficiencies. The denitrification process was enhanced at the higher COD/N ratio of 15/1 as compared to 5/1 by around 10–40% in all the unplanted CWH groups (Tables 2 and A3). Thus, as expected, increasing the COD/N ratio from 5/1 to 15/1 enhanced denitrification. Compared to cocopeat and mineral wool, the CWHs with pumice showed around 35% higher denitrification efficiency at COD/N of 5/1 and 50–60% at COD/N of 15/1. Cocopeat and mineral wool showed low and similar denitrification of around 10–20% and 25–40% at COD/N of 5/1 and COD/N of 15/1, respectively (Tables 2 and A3). This finding was further supported by a *t*-test conducted between the data for the two COD/N ratios and the three different substrata. Achieved p values of less than 0.05 in all the test groups explained that the observed higher denitrification at COD/N of 15/1 and the different quantity of denitrification achieved in those three different substrata were also statistically significant (Table A4).

### 3.2.3. Nitrification and Denitrification in CWH

The ability of CWH to facilitate both nitrification and denitrification processes is noteworthy and valuable for application. Nitrification is considered a limiting step in the N removal in traditional CWs due to insufficient aerobic conditions and retention time [17,31]. Denitrification on the other hand is limited by either a lack of organic C and/or the absence of anoxic conditions in the CWs [12]. To achieve both aerobic and anaerobic conditions simultaneously in a single CW, especially at a higher COD/N ratio, a certain degree of system heterogeneity is required, where aerobic zones (i.e., in the partly water-unsaturated pore space between the substratum grains, supporting nitrification) and anaerobic zones (i.e., inside the pores of substrata and inside the biofilms on the substrata, supporting denitrification) can co-exist.

The aerobic degradation of COD and nitrification occur in the aerobic parts of CW, but at high COD/N ratios, the DO concentrations could be reduced so far that the mesocosm would become fully anaerobic. This would lead to limiting nitrification, and limited production of $NO_3^-$-N and thus a decreased TN removal [32]. However, in this study, this did not occur: significant nitrification was observed in all systems at both studied COD/N ratios. Apparently, the CWH provided sufficient aerobic conditions, at least partially by an

influx of atmospheric oxygen into the CWH matrix, most likely through the pulse feeding in the vertical system [29,30].

Some degree of denitrification was also observed in all the CWH groups, indicating that anoxic conditions and COD were provided to drive the denitrification process or another microbiological path. Especially CWH-P exhibited significant denitrification ($89 \pm 5\%$) with a high COD/N ratio of 15/1 resulting in the TN effluent concentration which meets both end-use targets, namely the effluent requirements for reuse as irrigation water and for discharging into surface waters (Tables 1 and A3). This co-existence of anaerobic/anoxic and aerobic compartments within the filter bed of CWH is noteworthy, indicating the potential of such improved CWs to eliminate the limitations associated with removing N in the traditional CWs [33].

### 3.3. Effect of COD/N Ratio on N Removal

The effect of different COD/N ratios on denitrification was shown in this study and the finding was compared with other reported studies in the literature (Table 2). The denitrification process was enhanced by around 10–40% in all the unplanted groups of CWH at a COD/N of 15/1 compared to the COD/N of 5/1 (Table 2). A study observed significant nitrification (99%), TN removal (90%), and COD removal (96%) in intermittently aerated VFCWs treating domestic wastewater at an influent COD/TN ratio of 10/1 [17]. That study indicated that added extra carbon source enhanced denitrification process [17,34]; additionally, that study reported poor nitrification in non-aerated VFCWs due to lack of oxygen which led to lower COD and TN removal at high COD/N ratios and the authors suggested intermittent aeration combined with high COD/N ratio, as realized in CWH, as a way to enhance N removal in VFCWs.

In contrast to the previous findings, another study found the optimal COD/TN ratio of 5/1 to achieve high TN removal of 90% for the TN influent concentration of 40 mg/L and reported that the TN removal did not increase with an increasing COD/N ratio in horizontal flow CWs treating synthetic influent prepared by dissolving ammonium nitrate and glucose with tap water [20]. Similar to this study, another study reported a COD/TN ratio of 2.5–5 as the optimal range to achieve the most efficient TN removal in VFCWs treating synthetic sewage [35]. The found variations in the study results in terms of the optimal COD/N ratio for efficient TN removal could be ascribed to numerous factors, such as substrata type, plant species, organic C from substrata, design configurations, operational mode, and retention time of CWs [8,34].

The found variations in the literature for the optimal COD/N ratio to achieve higher TN removal in CWs also suggest that the COD/N ratio may not be the defining factor, but that COD concentration may be a determining parameter. At a high influent COD concentration of 750 mg/L (COD/N = 15/1), denitrification was enhanced compared with a low COD concentration of 250 mg/L. However, complete denitrification was not achieved even at the favorable COD/N of this study with COD of 750 mg/L. Therefore, COD might be a limiting factor for complete denitrification.

The addition of extra COD creates anoxic conditions which favor the denitrification process. Oxygen consumption for the oxidation of the additional COD could deplete the available oxygen in CWH, possibly in the bottom compartments or the inner spaces of pores and biofilms in or on the substratum grains of CWH where penetration of atmospheric oxygen might be limited, but where organic molecules can diffuse into. The anoxic conditions created due to the lack of oxygen could favor denitrification [17,34]. However, the addition of extra COD above the threshold could limit the nitrification process as oxygen availability could become limited due to the oxidation of COD, which did not occur in this study. Such oxygen depletion would thus limit the production of $NO_3{}^- $-N and eventually denitrification process. Additionally, it is important to remember that a threshold COD concentration is required to support denitrification [20].

### 3.4. Effect of Substrata on N Removal

Substrata type influenced the overall N removal in CWH. Similar nitrification was observed in CWH with cocopeat, mineral wool, and pumice (Table 2). All three substrata are highly porous and therefore, they could allow oxygen diffusion and provide aerobic conditions for nitrification to occur, which was further enhanced by the design of mesocosm CWH. Furthermore, the substrata possess reactive surfaces which could support microbial attachment, resulting in significant nitrification (Table A1) [32]. However, with regards to denitrification, CWH with pumice exhibited higher denitrification compared to cocopeat and mineral wool, accounting for 35% and 60% of denitrification at COD/N of 5/1 and 15/1, respectively (Table A3).

The difference in the composition of the three substrata and the filling (how the substrata are packed) in the filter bed may have influenced their performance in terms of denitrification. Considering the bulk characteristics of the three substrata, cocopeat, and mineral wool were uniformly packed in the filter bed of CWH. This could potentially lead to surface blockage of substrata via the formation of biofilms or creating more anaerobic conditions, resulting in low TN removal as reported in other studies [32,36]. However, non-uniform packing of pumice in CWH due to its particle size distribution varying from fine to coarse may have favored the formation of the aerobic and anaerobic niches and reduced the potential of surface blockage due to biofilms, resulting in less effect on TN removal [15,32,37]. Furthermore, among the three substrata, pumice has the highest surface area (Table A1) and therefore more available external and internal pore space for biofilm growth which together with the less potential of surface blockage may have positively contributed to better TN removal in pumice compared to cocopeat and mineral wool. In addition, non-uniform packing of pumice particles may have also resulted in low hydraulic conductivity and therefore higher retention time. Given sufficient time for oxygen to be fully consumed in pores and biofilms apparently favored the creation of anaerobic conditions for the denitrification process to occur and thus enhance overall TN removal. Other physicochemical characteristics of substrata, such as surface charge could have influenced TN removal. For example, the cocopeat surface is negatively charged which may have limited microbial adhesion and biofilm formation on the surface, resulting in low denitrification compared to pumice and mineral wool which are neutral (Table 2).

### 3.5. Role of Plant on N Removal

Syngonium plant enhanced TN removal in all the planted groups in both COD/N ratios (Table 1). At COD/N of 5/1 in unplanted groups, mineral wool, and cocopeat showed very low TN removal of less than 7%. By adding the plants, 50% additional TN was removed, mostly likely by phytoremediation. CWH with pumice showed on average 35% TN removal without plants, and by adding plants, 50% additional TN removal was achieved. TN removal was increased by around 10–20% from COD/N of 5/1 to COD/N of 15/1 for planted groups (Table 1) which may be due to the increases in N requirement of Syngonium from the initial growth stage (during COD/N of 5/1) to later growth stages (during COD/N of 15/1).

Generally, plants grown in acidic soil conditions prefer N in the form of $NH_4^+$-N and those in alkaline soil conditions prefer N in the form of $NO_3^-$-N [38]. Concerning the pH of CWH substrata, cocopeat has a naturally acidic pH whereas pumice and mineral wool have a neutral pH (Table A1). However, all the CWH groups were fed with a P buffer solution to maintain a neutral pH in the system to facilitate nitrification and denitrification processes. Therefore, the preferable form of N for Syngonium ($NH_4^+$-N or $NO_3^-$-N) in the CWH filter bed could not be deduced in this study. However, it is important to gain specific knowledge concerning the N requirement of Syngonium in CWH substrata in order to optimize the plant's contribution to N removal.

### 3.6. Observed Total Phosphorous (TP) Removal in CWH

In addition to N removal, this study examined TP removal in CWH. The prepared influent for CWH contained 10 mM of P buffer of $Na_2HPO_4.2H_2O$ and $KH_2PO_4$ in order to maintain the optimal pH of around 7 for the nitrification and denitrification process to occur (Figure A1). During the operation, the absolute P removal of $44 \pm 19$ mg/L was observed in CWH-P. However, the influent P concentration was around 269 mg/L which is an unrealistic influent concentration for urban wastewater treatment. Therefore, it is suggested to study the potential of CWH, particularly CWH-P to remove TP at more environmentally relevant starting concentrations.

### 3.7. Design of CWH for N Removal

The effluent of CWH is mainly targeted for reuse in agriculture as fertigation water. When this reuse option is not feasible, the effluent could be discharged into surface waters if regulations are met. From this study, it is apparent that at high COD/N (15/1), both planted and unplanted groups of pumice (CWH-PS and CWH-P) met the reuse requirements of irrigation water for TN (Table 1). However, the produced effluents of cocopeat and mineral wool systems exceeded the irrigation standard for TN at both COD/N ratios. When considering the discharge into surface waters option, CWH-PS, CWH-P, and CWH-MS met the effluent requirements for TN at COD/N: 15/1; other groups exceeded the standards (Table 1). In summary, TN removal in cocopeat and mineral wool needs to be improved in order to comply with the irrigation standards.

To enhance the ability of CWH to facilitate both nitrification and denitrification processes, certain conditions have to be sustained at CWH. Concerning nitrification, as observed, complete or significant nitrification could be achieved due to the vertical design and pulse feeding which provides the required aerobic conditions for the nitrification process to occur. Therefore, to improve the overall TN removal in CWH, the denitrification process needs to be further enhanced by providing the required conditions: an anoxic environment, neutral pH, sufficient retention time, and COD availability.

As discussed above, an addition of extra COD above 750 mg/L or increasing the COD/N ratio above 15/1 may not be a solution to improve denitrification in CWH concerning a post-treatment process. If CWH is considered to treat the raw wastewater, for example, municipal wastewater, high COD could be retained in the influent by adding a sedimentation tank as the pre-treatment step where COD and TN are not removed but solid particles to prevent clogging in CWH. It is also important to consider that the COD used in this study is methanol which is easily biodegradable. However, raw wastewater may contain different kinds of COD, for example, non-biodegradable COD or slowly biodegradable COD which cannot be easily used by microorganisms for biosynthesis or energy production [39]. Therefore, instead of increasing COD concentration in wastewater with extra COD, other approaches focusing on the CWH design and operational parameters to enhance denitrification in different matrices of CWH could be considered.

The denitrification process can be enhanced by providing anoxic conditions with sufficient availability of COD. For this purpose, CWH design and operational parameters could be altered to reduce the oxygen supply to the system. By increasing the frequency of pulse feeding to CWH, the available pore spaces in matrices could be better occupied with water than air, which could reduce the oxygen supply to the filter bed of CWH. Additionally, by increasing the depth of the CWH bed and thus increasing the depth-to-surface area ratio, the available surface for oxygen penetration to the filter bed could be reduced. Further study is required to better understand the DO profile of CWH to improve the nitrification and denitrification processes. In addition, model-based studies incorporating the major N removal processes in CWH, such as nitrification, denitrification, and phytoremediation, and the suggested changes in the design and operational configurations need to be conducted. This would lead to the development of effective models to predict the performance of CWH and optimize the performance for producing effluent that complies with discharge or reuse standards [40].

## 4. Conclusions

The study was conducted to understand the effect of the influent COD/N ratio of 5/1 and 15/1, substrata type, and the presence and activity of plants on N removal or conservation in CWH. The observed extensive nitrification of above 90% in all the CWH groups indicates that different COD/N ratios and substrata types did not limit nor enhance the nitrification process. Vertical design and pulse feeding regime provided sufficient atmospheric oxygen into the CWH to facilitate this extensive nitrification. Denitrification was enhanced in CWH when changing COD/N from 5/1 to 15/1. This confirmed that the addition of an external carbon source such as methanol increases the overall N removal in CWH. It was also found in this study that substrata type influenced the N removal in CWH. CWH with pumice exhibited higher denitrification compared to cocopeat and mineral wool at both COD/N ratios. Furthermore, Syngonium plants enhanced the TN removal, in addition to the nitrification and denitrification processes in the CWH matrix.

To maintain nitrification and denitrification processes, the coexistence of aerobic and anaerobic conditions needs to be sustained in CWH which could be attained by increasing the frequency of pulse feeding, the depth-to-surface area ratio of CWH, and the choice of the substratum. A clear understanding of the DO profile of CWH would help to improve the N removal processes in CWH. While considering the reuse options of the treated water, CWH, particularly, CWH with pumice could produce effluent which meets effluent quality requirements, for both discharging into surface waters and irrigation water, which is valuable for future applications, especially in water-scarce areas.

**Author Contributions:** The authors contributed to this publication as follows: E.S.: conceptualization, methodology, formal analysis, writing—original draft, writing—review and editing, visualization; N.B.S.: conceptualization, methodology, writing—review and editing, visualization, supervision; H.H.M.R.: conceptualization, writing—review and editing, supervision; K.K.-R.: conceptualization, methodology, writing—review and editing, visualization, supervision. All authors have read and agreed to the published version of the manuscript.

**Funding:** This research is a part of the LOTUS-HR project which is an Indian–Dutch consortium, funded by the TTW (Toegepaste en Technische Wetenschappen)-NWO (Netherlands Organisation for Scientific Research), DBT (Department of Biotechnology) India and Environmental Technology department, Wageningen University and Research, the Netherlands.

**Data Availability Statement:** The data presented in this study are available on request from the corresponding author. The data would be publicly available upon completion of the PhD research of the first author.

**Acknowledgments:** The authors would like to thank Hennie Dorland from Wageningen University and Research, the Netherlands for his assistance in the technical design of the experimental set-up.

**Conflicts of Interest:** The authors declare no conflict of interest. The funders had no role in the design of the study; in the collection, analyses, or interpretation of data; in the writing of the manuscript; or in the decision to publish the results.

## Appendix A

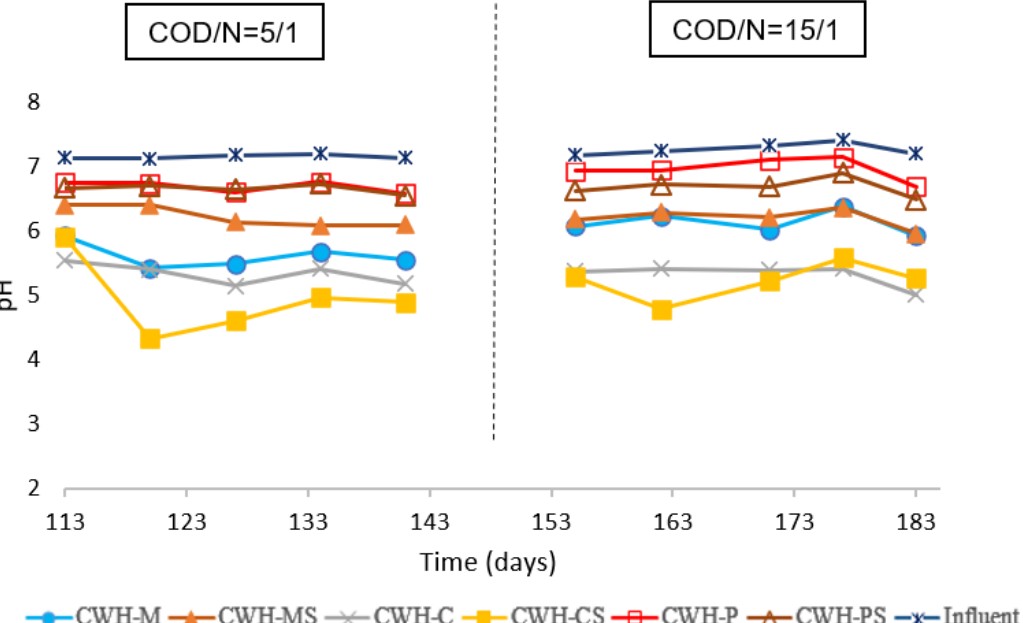

**Figure A1.** Influent and effluent pH of all the studied CWH. CWH-MS and CWH-M indicate mineral wool + syngonium and mineral wool unplanted groups respectively. CWH-CS and CWH-C indicate cocopeat + syngonium and cocopeat unplanted groups respectively. CWH-PS and CWH-P indicate pumice + syngonium and pumice unplanted groups respectively.

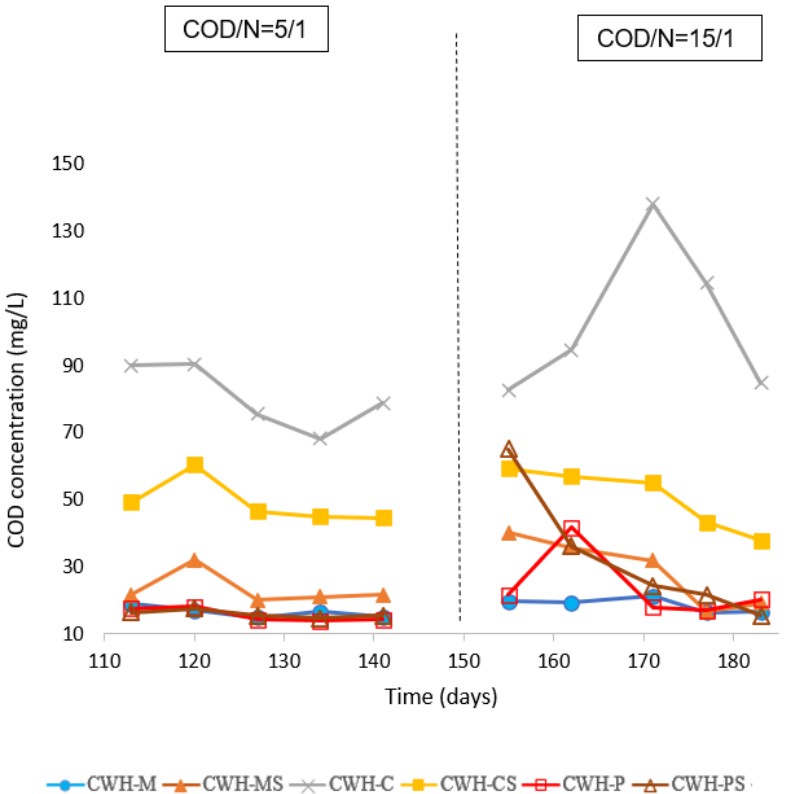

**Figure A2.** COD concentration in all the studied CWH effluents over the experimental period.

**Table A1.** Physicochemical characteristics of the substrata [15].

|  | Mineral Wool | Cocopeat | Pumice |
|---|---|---|---|
| pH in water | 7.7 | 6 | 7 |
| Organic matter (%) | 3 | 90–100 | <1 |
| Surface area ($m^2$/g) | $0.5 \pm 0.5$ | $1.8 \pm 0.7$ | $2.4 \pm 0.$ |
| Porosity (%) | 97 | 97 | 72 |

**Table A2.** Estimated nitrification (%) in the unplanted CWH over the experimental period.

| COD/N Ratio of Influent | 5/1 | | | | | 15/1 | | | | |
|---|---|---|---|---|---|---|---|---|---|---|
| Days | 113 | 120 | 127 | 134 | 141 | 155 | 162 | 171 | 177 | 183 |
| CWH-M | 95 | 96 | 97 | 96 | 96 | 98 | 99 | 97 | 99 | 99 |
| CWH-C | 93 | 98 | 87 | 96 | 99 | 100 | 100 | 98 | 99 | 100 |
| CWH-P | 94 | 87 | 88 | 89 | 91 | 96 | 93 | 96 | 95 | 92 |

**Table A3.** Estimated denitrification (%) in the unplanted CWH over the experimental period.

| COD/N Ratio of Influent | 5/1 | | | | | 15/1 | | | | |
|---|---|---|---|---|---|---|---|---|---|---|
| Days | 113 | 120 | 127 | 134 | 141 | 155 | 162 | 171 | 177 | 183 |
| CWH-M | 9 | 5 | 15 | 10 | 16 | 40 | 38 | 29 | 40 | 39 |
| CWH-C | 10 | 9 | 34 | 12 | 16 | 33 | 26 | 25 | 23 | 21 |
| CWH-P | 56 | 39 | 44 | 54 | 50 | 95 | 80 | 90 | 91 | 91 |

**Table A4.** *p*-values obtained from *t*-test conducted.

|  | *p*-Value | |
|---|---|---|
|  | Nitrification | Denitrification |
| CWH-M | 0.15 | 0.00 |
| CWH-C | 0.19 | 0.02 |
| CWH-P | 0.13 | 0.00 |

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
