# Peer review of "Controlling Nitrogen Removal Processes in Improved Vertical Flow Constructed Wetland with Hydroponic Materials: Effect of Influent COD/N Ratios"

_water, doi:10.3390/w15061074_

Round 1

Reviewer 1 Report

The manuscript raises the problem of effect different ratio influence (COD/N) on nitrogen removal in constructed wetland with hydroponic materials (CWH). Authors conducted an experiment in which three different substrata as pumice, cocopeat and mineral wool were observed. In each mesocosm with selected substrata, Synogium as vegetation was planted. Experiment were conducted for 28 days with synthetic influent used. Conducted research  were focused on COD, NH4-N, NO3-N, TN  analysis. Based on these laboratory tests Authors calculated removal efficiency for total nitrogen.

The article is original work in its character and in Reviewer opinion corresponds precisely with Journal content. The manuscript is properly designed, it is divided on main chapters: Introduction, Materials and Methods, Results and Discussion and Conclusions. The paper is prepared in plain language. The Abstract content allows the Readers assess the main aim and expected results of the research. The tables and graphs used in text are correct. More than 40% of cited references are older than 5 years. For such journal as “Water” is, cited source should be recent as possible.

The main doubts, in Reviver opinion, are raised by the content of table 2 (in main text) and table A2 and A3 (appendix). There are no clarification of Measured nitrification/nitrified NH4-N as well Estimated denitrification/denitrified NO3-N mentioned in table 2.  It seems to be a concentration difference in inflow and outflows. Similar comments are for value in table 2A and 3A. Authors described them as measured nitrification (in %) in fact there are removal efficiency. Nitrification rate we can estimated using value of influence, concentration of ammonium and total nitrogen, volume of reactor as vell as concentration volatile of suspended solid  in reactor. This part of manuscript according to Reviver opinion need to be clarified.

Constructed wetland technology with hydroponic material is a very interesting issue and this experiment regarding to hydroponic material using seems like a new idea.

Final conclusion: After reading the manuscript, in its present form, level of advancement, reliability and scientific value, the manuscript can be published in Water after taking into account major comments and editorial.

More detailed comments are listed below:

Chapter 2. Material and Methods:

1.       please specify the hydraulic load of the system, HRT and pollution load of the surface,

2.       whether the authors measured the velocity of the wastewater  flow through the system,

3.       please specify, in acclimatized period mesocosms were rinsed with water or wastewater?

4.       please specify, whether the dose of wastewater was fed to the bed once?

5.       please specify, whether the outflow  sample was average sample or instant sample?

6.       please explain why Authors chose the Synogium plant?

Chapter 3. Results and Discussion:

1.       Please specify calculation method of Measured nitrification/nitrified NH4-N as well Estimated denitrification/denitrified NO3-N mentioned in table 2 as well as measured nitrification (in %) mentioned in table 2A and 3A.

Reviewer 2 Report

The topic discussed in the manuscript is aligned with the journal's aim and scope and it is a timely topic to discuss. However, there are concerns about the significance of the content and the results presented in the manuscript. I think there are a number of missing figures and tables in the manuscript i.e. A1, A2, A3, A4,... which I can not see in the submitted manuscript. Authors should refer to the below points to revise the manuscript and provide the missing figures in the revised manuscript. 

The abstract should give some information about the 'Improved vertical flow constructed wetland' - is this done on a laboratory scale? is this done in a real field? what conditions were used for the test? info to help readers better understand the study. 

The introduction is supposed to conduct a literature review and gap analysis which is currently missing. The study's significance is not highlighted, for example, we know the carcinogenic and non-carcinogenic risks of nitrate for human health and this should be used to highlight the significance of removing N and P (10.1016/j.jclepro.2022.132432) L27.  
I think the use of nature-based solutions such as constructed wetlands should be justified in terms of the necessity of reducing emissions from traditional wastewater treatment plants (10.1007/s11356-020-08277-3; 10.2166/wst.2020.220). 
Recent literature on N removal methods with CW and biodegradation should be briefly mentioned so that the readers can refer to the recently published work. similarly, literature on P removal and constructed wetlands should be mentioned. L77-89 should include references from recent studies (10.1016/j.jtice.2021.01.030). 
Although this study is about improving the design and performance of CWs you didn't mention anything about model-informed design, how sensitive CWs are to very small changes, and how models can be used to inform detailed design and operations (10.1016/j.ecoleng.2022.106702). 
Material and Methods: Where is Table A1? why this table is not provided in this section? Figure A2 is not provided, I recommend putting this figure in the manuscript text as Figure 1.
L140-142 - if you can provide a picture of the test set up would be useful for the readers. 
I think it will be beneficial for the readers to see a flowchart or schematic summarising the key processes described in sections 2.1 - 2.5. Think about presenting the method in one figure/flowchart. 
Also, I think experimental conditions can be summarized in a table so readers can easily understand the conditions tested in your study.  
Results: The manuscript that I received, has only two tables in the results section and no figures. Where is figure A3? Also, both tables can benefit from formatting enhancement. Where is Table A2? Where is Table A4? 
Overall, the description of the results and structure of the results section are good but missing tables and figures make it difficult to properly assess the results. There are lots of descriptions of the results but not many discussions on how the results relate to the existing findings or what is the impact of the findings. you need to better highlight the significance of your results. 
Conclusions: mostly fine, you can better highlight the key findings and new information generated by your study. 

Reviewer 3 Report

Dear authors, after reading your manuscript, I have the following commenters:

1. It would be good to add technological schemes for wastewater treatment in the work. It would be clear to the readers that the process of reducing the nitrogen concentration by nitrification and denitrification occur sequentially, not simultaneously.

In the work, it is necessary to write the practical value of the results obtained, which can be implemented in practice.

2. It is known that the processes of nitrification and denitrification are very sensitive to the concentration of oxygen in the water. There is no information about this in the work.

3. line 282 - it is known

4. In the conclusions, the authors write that with an increase in the SOD/N ratio, an increase in the concentration of oxygen increases the intensity of nitrification, so this is known! The type of organic matter affects the nitrification process, so this is also known. Methanol is an easily oxidized organic substance, and yes, it is a good carbon source for nitrifying bacteria. And how do hard-to-oxidize organic substances affect?

Unfortunately, I did not see any innovations or scientific novelty in this work

Round 2

Reviewer 2 Report

Authors have revised the manuscript and improved the content which is very good. However, some of the point raised in the previous round of revision still remains unaddressed. See below comments and revise accordingly. 

This time I could see the appendix file and assess the content. In my opinion there is no need for these figures/tables to remain as appendix and they should be added to the manuscript text. Your manuscript has only two tables and no figures in the current format, which does not conform with the usual norms. 

I still think the literature/ discussions of the results can briefly mention the necessity of using model-informed studies to to improve the design and performance of CWs (10.1016/j.ecoleng.2022.106702; and  10.1016/j.jwpe.2020.101411) and also the necessity of developing appropriate models that can capture all the relevant processes. Expectation here is not for you to conduct model-based optimization but to raise the necessity of this and the available models. 

The manuscript can be considered for publication following a revision based on above comments. 

Author Response

We thank the reviewer for the feedback on our improved version of the manuscript. 

Regarding the point 1, Now we moved Figure A1 and Figure A2 (CWH set up pictures) to the manuscript as we also agree that the previous version did not have any figures so the added pictures would be a nice inclusion. However, we do not still think the COD and pH figures and the tables are important to be included in the manuscript. For example, if we take Table A2 and A3, they are removal efficiency values. These values were helpful to make comparison between different CWH system for their performance. However, these two tables are presented in the manuscript as Table 2 with the estimated nitrification and denitrification values in mg/L. Basically, Table 2 and Table A1 and A3 present the same information but in different ways. Therefore, we think those tables in the appendix are not crucial to be put in the manuscript.

Regarding point 2, now we added the following text at the end of discussion part where we talk about improvements of the system. We also used the article referred by the reviewer.

"In addition, model based studies incorporating the major N removal processes in CWH, such as nitrification, denitrification and phytoremediation and the suggested changes in the design and operational configurations need to be conducted. This would lead to the development of effective models to predict the performance of CWH and optimize the performance for producing effluent that complies with discharge or reuse standards [39]."

Reviewer 3 Report

I agree such version of manuscript

Author Response

We thank the reviewer for the positive response.

Round 3

Reviewer 2 Report

The authors have revised and improved the manuscript. The manuscript need proofreading and formatting during the production.